# The assessment of psychometric properties of childbirth violence questionnaire in Iranian women

**Molouk Jaafarpour[1], Ziba Taghizadeh[2], Abbas Ebadi**  **[3,4], Fatemeh Abbaszadeh[5,6], Fatemeh Najafi[7], Ashraf Direkvand-Moghaddam** [8]*

**1** Department of Midwifery, School of Nursing and Midwifery, Ilam University of Medical sciences, **2** Nursing and Midwifery Care Research Center, Tehran University of Medical Sciences, Tehran, Iran, **3** Behavioral Sciences Research Center, Life Style Institute, Baqiyatallah University of Medical Sciences, Tehran, Iran, **4** Nursing Faculty, Baqiyatallah University of Medical Sciences, Tehran, Iran, **5** Trauma Nursing Research Center, Kashan University of Medical Sciences, Kashan, Iran, **6** Department of Midwifery, School of Nursing and Midwifery, Kashan University of Medical Sciences, Ilam, Iran, **7** Department of Nursing, School of Nursing and Midwifery, Ilam University of Medical Sciences, Ilam, Iran, **8** Department of Midwifery, School of Nursing and Midwifery, Ilam University of Medical sciences

* direkvand-a@medilam.ac.ir

## Abstract

### Background

Childbirth violence is a cultural and social issue with a complex and multidimensional nature that silently threatens women's health around the world. Thus, designing an instrument based on social norms and the cultural context of society is an essential need. The study aims to develop and validate a measurement instrument that assesses childbirth violence, focusing on the experiences of women during childbirth.

### Methods

The study utilized a mixed-methods approach which is a combination of qualitative and quantitative research methodologies and evaluated a new tool for assessing violence during childbirth. It included three main phases: Instrument Development, Validity, and Reliability. The instrument's validity was tested through exploratory factor analysis (EFA). To assess sampling adequacy, the Kaiser-Meyer-Olkin index. Multiple extraction and rotation techniques were employed to determine the optimal questionnaire structure.

### Results

After completing the qualitative phase, the research team focused on two key themes: the violation of pregnant mothers' rights within the healthcare system and societal factors contributing to violence against pregnant mothers. 99 items were identified based on participants' perspectives and interviews. Existing questionnaires

**Data availability statement:** All relevant data are within the manuscript and its Supporting information files

**Funding:** The author(s) received no specific funding for this work.

**Competing interests:** The authors have declared that no competing interests exist.

**Abbreviations:** CBVQ, ChildBirth Violence Questionnaire; CVI, Content Validity Index; CVR, Content Validity Ratio; EFA, Exploratory Factor Analysis; KMO, Kaiser-Meyer-Olkin (index for sampling adequacy); ICC, Intraclass Correlation Coefficient; SEM, Standard Error of Measurement; RMC, Respectful Maternity Care.

on respectful maternal care were reviewed, but none specifically targeted childbirth violence. A comprehensive pool of 108 items was refined to a preliminary questionnaire with 65 items for psychometric testing. EFA, based on maximum likelihood and varimax rotation, concealed that questionnaire contains four factors including; psychological and physical abuse, non-professional care, delay in offering care and lack of family support. These factors explained 63.276% of the total variance. Cronbach's alpha (0.88–0.90), (ICC = 0.95 (and (SEM = 3.2), confirmed the sufficient reliability of the Childbirth Violence Questionnaire with 34 items (CBVQ-34).

## Conclusions

The CBVQ is a unique tool to assess childbirth violence through mothers' experiences during natural birth or emergency cesarean sections. It focuses on violence from care teams, the health system, and families, unlike existing tools that mainly highlight the care team's relationship with the mother. The CBVQ also considers the health system's structures and family support, distinguishing it from other tools.

## Background

Childbirth is one of the critical stages in women's lives that can have profound effects on their physical and mental health [1].Advances in medical knowledge and technology have shifted childbirth to hospitals to reduce mortality rates, making it a public event. This has portrayed childbirth as a condition needing medical intervention, diminishing women's role during labor. Health professionals, especially obstetricians and midwives, dominate decision-making processes, leading to a loss of women's autonomy [2].

Obstetric violence (OV) refers to inappropriate or disrespectful treatment given by medical staff to women during pregnancy, delivery, and postpartum [3]. Mistreatment of women during childbirth is a significant global issue, emphasizing its widespread occurrence and the cultural tolerance that allows it to persist. A study in Kisangani, DRC, found that 21.63% of women were denied family support during delivery, and 58.65% experienced non-consensual vaginal examinations [4]. In a study of 256 women in Odisha, India, 32.4% experienced obstetric violence during childbirth [5]. This violence is often supported by existing policies and procedures that favor paternalistic approaches in healthcare settings [6].OV is described as a form of gender-based violence that targets women during childbirth, violating their human rights and undermining the delivery of respectful maternity care (RMC) [7].

It points out that labor and delivery nurses are the healthcare professionals who spend the most time with patients during childbirth. Their unique position places them in a critical role where they can either prevent or contribute to obstetric violence [6]. Disrespect and abuse during childbirth are significant concerns in Iran. A cross-sectional study conducted in Tehran revealed that 68.3% of mothers experienced at least one form of disrespect or abuse during childbirth [8]. Efforts to reduce childbirth violence and enhance the birth experience have gained significant attention in

recent years, focusing on humanization and RMC. These initiatives aim to empower women during childbirth, ensuring they receive dignified and supportive care. Efforts include educating nursing staff on obstetric violence, promoting humanized care, respecting women's choices, and prioritizing their well-being to enhance the childbirth experience and combat obstetric violence [9].

Efforts include RMC training for providers, open maternity days, clinical checklists, wall posters, and user feedback to reduce mistreatment and enhance birth experiences [10].

Based on the results of a study, efforts to reduce childbirth violence and enhance the birth experience include humanized care interventions, allowing choices like eating, drinking, and position selection, but require functional facilities and supportive healthcare providers [11]. So far, tools have been designed to investigate violence in different reproductive ages [1,12]. A study focuses on translating and validating the Persian version of the Pregnancy and Childbirth Questionnaire (PCQ) to measure Iranian women's satisfaction with prenatal and intrapartum care quality. The results introduced the PCQ's Persian version as a valid and reliable instrument for measuring satisfaction with prenatal and intrapartum care quality in Iranian women [11]. Another study assesses the psychometric properties of the Disrespect and Abuse Questionnaire in Iranian postpartum women, confirming its validity for evaluating the lack of respectful maternity care [1]. The developed tool is deemed essential for measuring childbirth violence, addressing a significant gap in existing research and practice. By reviewing the available sources, the researchers found that, there was no specific, comprehensive, and multidimensional questionnaire designed to assess childbirth violence in Iranian women. This absence highlighted a significant gap in both research and practice, necessitating the creation of a tailored instrument to address this issue.

Also, childbirth violence is a complex cultural and social issue that varies across different societies. The researchers recognized the need for an instrument that reflects the unique experiences and challenges faced by Iranian women during childbirth. In the present study, the design and psychometrics of tools measuring childbirth violence in Iran have been explored. The development of these tools is crucial for understanding and addressing the experiences of women during childbirth.

## Materials and methods

This research was conducted from August to December 2019 and employed a blend of quantitative and qualitative methodologies to design and conduct a psychometric evaluation of the Iranian instrument intended for assessing violence during childbirth. The process of accessing the instrument was organized into three primary phases: Instrument Development, Instrument Validity, and Instrument Reliability.

### Instrument development

Creswell indicates that in a mixed-methods study, following the qualitative phase and the elucidation of the intended concept, the research team is positioned to determine which themes will be utilized in the development of the measurement instrument [12]. To finalize the items, a review of various sites and databases, including Google Scholar, Pubmed, ProQuest, Scopus, Science Direct, Springer, as well as dedicated sites from the WHO and CDC, was conducted for the period from 1990 to 2020. The search for available questionnaires utilized keywords such as Childbirth Violence, Obstetrics, Disrespect and Abuse, Mistreatment, Tool, Questionnaire, and Instrument.

### Instrument validity

The second phase of the study involved the assessment of instrument validity. The validation process included face validity, content validity, and factor analysis.

### Face validity

The face validity was assessed through a combination of qualitative and quantitative approaches. A qualitative survey was administered in person to a group of 15 women. Inclusion criteria consisted of women of reproductive age (18–49)

who had experienced childbirth (either vaginal delivery or elective cesarean section) within the past three years in various hospital settings (public, semi-public, and private). Participants were required to speak Persian and be capable of communicating and providing information. Women who had physical or mental illnesses affecting pregnancy and childbirth outcomes were excluded from the study. The investigation focused on the challenges associated with comprehending certain phrases and words, as well as the potential for misinterpretations and inadequacies in word meanings. Subsequently, the items were revised by the feedback provided by this group.

Then the writing of 5 items was changed due to the difficulty in understanding its meaning. Subsequently, to assess the significance of each item, a quantitative face validity analysis was conducted by evaluating the item's impact score. At this stage, 15 women were asked to give their opinions about the importance of the items based on a Likert scale of completely important [5], somewhat important [4], moderately important [3], slightly important [2], and not important [1].

## Content validity

Content validity represents a crucial aspect of instrument validity, relying on the perspectives of a panel of experts [13]. To assess content validity, twelve experts and university professors specializing in reproductive health, midwifery, psychology, sociology, and nursing were consulted to evaluate the appropriateness and placement of phrases, providing essential feedback. Regarding qualitative content validity, the written structure of five items was revised based on the experts' recommendations. Additionally, twelve overlapping items were consolidated, and two items were eliminated. For the evaluation of quantitative content validity, the Content Validity Ratio (CVR) and Content Validity Index (CVI) were employed. Lawshe has proposed a quantitative model for assessing content validity, wherein a questionnaire is distributed to experts who evaluate the relevance of its items [14]. The experts' responses are recorded using a 3-point Likert scale, categorized as "it is necessary," "it is useful but not necessary," and "it is not necessary." Subsequently, these responses are quantified to calculate the CVR.

During the quantitative content validity phase, the questionnaire was distributed to 20 faculty members with expertise in midwifery care within the delivery department. Out of these, 12 individuals completed the questionnaire. The CVR serves as a statistical measure utilized to determine the rejection or acceptance of specific items. In the CVR assessment, ten experts were invited to evaluate each item on a scale from 1 to 3, corresponding to the categories of "not necessary," "useful but not essential," and "essential." A higher level of content validity indicates a greater consensus among the experts. The formula for calculating CVR, as proposed by Lawshe in 1975, is expressed as $CVR = (N_e - N/2)/(N/2)$, where $N_e$ represents the number of panelists who rated an item as "essential," and N denotes the total number of panelists. The numerical value of the content validity ratio is referenced from the Lawshe table. The evaluation of each item was conducted according to the following criteria:

- If fewer than half of the panelists deemed an item "essential," the CVR was considered negative.

- If exactly half of the panelists rated an item as "essential" while the other half did not, the CVR equated to 0.

- If all panelists rated an item as "essential," the CVR was calculated as 1.00 (though it is adjusted to 0.99 for practical purposes). In cases where the number of panelists rating an item as "essential" exceeded half but did not reach the total, the CVR fell within the range of 0 to 0.99.

To assess the CVI, 10 experts from the fields of reproductive health, psychology, sociology, and nursing were invited to evaluate the relevance of each item about the intended framework. The relevance of each item was rated by specialists using a 4-point Likert scale, with the following designations: "completely relevant" [4], relevant [3], "somewhat relevant" [2], and "not relevant" [1].

The CVI score was calculated for the item (Item Content Validity Index-I-CVI) and the entire instrument (Scale Content Validity Index-S-CVI). Evaluating the S-CVI is one of the basic steps in increasing and improving the construct validity of

the instrument, which is carried out by two methods: S-CVI/UA and S-CVI/Ave. In this research, the S-CVI/Ave method was used to evaluate the validity of the entire content of the instrument, and this rate was 0.9. The acceptance criterion of S-CVI among the experts is 0.90 as an excellent criterion and a numerical value of 0.80 is the minimum acceptable index of validity of the entire content of the tool.

To determine the I-CVI, Polit and Beck introduced a modified kappa statistic, which serves as a measure of agreement among evaluators concerning the relevance of items. They assert that the interpretation of the modified Kappa coefficient is as follows: a score ranging from 0.40 to 0.59 indicates poor agreement, a score from 0.60 to 0.74 signifies good agreement, and a score of 0.74 or higher is considered desirable.

## Preliminary reliability results before determining construct validity (item analysis)

The tool's reliability was checked by internal consistency and by calculating Cronbach's alpha coefficient for each item (above 0.7 desirability criterion) and the relationship of each item with the whole instrument (above 0.3 desirability criterion) in a sample of 30 mothers.

## Construct validity

The results of construct validity were assessed through exploratory factor analysis. Following the acquisition of the necessary permissions, a questionnaire was administered to 205 mothers who had inclusion and exclusion criteria mentioned in the face validity. The data collected was analyzes using SPSS software version 26.

To evaluate the suitability of the data for factor analysis, both the adequacy of sampling and the operability of the data were scrutinized. The Kaiser-Meyer-Olkin (KMO) index was employed to assess sampling adequacy, while Bartlett's Test of Sphericity was utilized to evaluate data operability.

## Factor extraction

The maximum likelihood method combined with varimax rotation was selected for the initial factor extraction process. To determine the number of factors in the childbirth violence questionnaire, both the eigenvalue criterion and the scree plot were employed. A criterion was established whereby factors with an eigenvalue exceeding 1, positioned above the horizontal line, and accounting for at least 50% of the variance of the intended concept were considered significant. To refine the item structure, multiple analyses were conducted utilizing various extraction and rotation techniques, ultimately identifying a four-factor structure as optimal.

## Instrument reliability

The assessment of reliability, internal consistency (measured by Cronbach's alpha), and stability (evaluated through the intra-cluster correlation coefficient and measurement standard error) was conducted using the test-retest method. For this evaluation, a group of 30 participants was instructed to complete the developed questionnaire, which was subsequently administered again under identical conditions after a period of 14 days. The intra cluster correlation coefficient [1] was computed utilizing SPSS software, employing a two-way mixed model with a 95% confidence interval for four constructs and all instruments. It is noteworthy that an ICC value exceeding 0.7 is considered acceptable.

## Ethics approval and consent to participate

The study followed strict ethical guidelines, approved by the Ethics Committee of Tehran University of Medical Sciences (Code: IR.TUMS.VCR.REC1398.046), and aligned with the Declaration of Helsinki. All participants gave informed consent after being told their identities would stay confidential. Before interviews began, researchers explained that no real names or personal details would be in any published findings. This point was also clearly outlined in the consent form. To protect privacy, participants were either assigned pseudonyms or referred to as "the participants" throughout the manuscript.

## Results

Upon concluding the qualitative phase, the research team opted to focus on the first two themes, namely, "the violation of the rights of pregnant mothers within the healthcare system" and "the societal factors contributing to violence against pregnant mothers" to inform the creation of the tool. The third theme, which pertains to negative health consequences, was not incorporated in this context. Drawing from the participants' perspectives on the concept of childbirth violence, along with direct quotations from the interviews, a total of 99 items were identified to create a questionnaire aimed at assessing childbirth violence. In this study, the inductive-comparison approach was employed to develop the tool. Review of various sites and databases revealed that there were no questionnaires specifically designed to assess childbirth violence. The findings were restricted to those measuring respectful maternal care, including the RMC scale, QRMCQ, and WP-RMC. Additionally, nine items were adapted from previous studies and questionnaires relevant to this area. Consequently, the researcher developed a comprehensive pool of 108 items. During several face-to-face and virtual (Skype) meetings, these items were examined and finally confirmed by the research team to ensure accuracy and to find any overlapping and duplicated items. After review and refinement by the research team, several similar items were merged or removed, and finally, a preliminary questionnaire with 65 items was formed and prepared for the start of psychometrics. After this stage, the designed primary tool was examined in the psychometric stages related to the quantitative part of the study. In the quantitative face validity assessment, the impact scores for all items exceeded 1.5, indicating acceptability. The lowest score recorded was 1.76, while the highest reached 5. During the face validity evaluation, no items were eliminated. Consequently, the tool proceeded to content validity with a total of 65 items. The questionnaire designed to assess childbirth violence progressed to the quantitative content validity phase, comprising 55 items following revisions made during the qualitative content validity phase. According to Lawshe's table (2014), the minimum acceptable CVR for 12 experts was set at 0. 56. As a result, 11 items were removed for scoring below 0. 56, leading to a refined tool of 44 items for evaluating the content validity index. Although Item No. 52 had a lower score of 0. 74, it was kept due to its importance and team agreement. The questionnaire, now with 44 items, moved on to further psychometric evaluation. In preliminary reliability testing, Cronbach's alpha was 0. 956, and three items were removed for having a correlation of less than 0. 3 with the overall instrument.

The KMO value of 0.777 obtained in this study indicated that the sampling was adequate for factor analysis. Additionally, Bartlett's test revealed a significant correlation among the items. Consequently, the findings from this phase confirmed that conducting factor analysis based on the correlation matrix was warranted.

According to the eigenvalue analysis, four factors exhibited eigenvalues greater than 1, while subsequent factors displayed nearly equivalent levels. Collectively, these four factors accounted for 63.276% of the total variance across 34 items. Table 1 illustrates that these four primary factors constitute the dimensions of childbirth violence.

The Scree Plot shows that 4 factors have a suitable distance and have an eigenvalue greater than 1, and the changes in the eigenvalue decrease from the fourth factor onwards. Therefore, it indicates that the first 4 factors are the main context of the items. The Scree Plot of the questionnaire of the Iranian instrument for assessing violence during childbirth is presented in Fig 1.

Consequently, the questionnaire was completed with a total of 34 items. The phases involved in the reduction of items, from the creation of the initial item pool to the completion of the final questionnaire version is illustrated in Table 2.

Subsequently, the factors were designated according to the established terminology associated with each factor and their alignment with the concept of childbirth violence identified during the qualitative phase. A minimum acceptable factor loading value of 0.3 was established. Items exhibiting the highest correlations were grouped into a single factor. The first factor comprised 14 questions, accounting for 26.215% of the variance; the second factor also included 14 questions, explaining 22.87% of the variance; the third factor contained 3 questions, representing 7.28% of the variance; and the fourth factor, with 3 questions, elucidated 6.913% of the variance. These factors were designated as "psychological and physical abuse"," non-professional care", "delay in offering care" and "lack of family support", respectively (Table 3).

**Table 1. The total variance explained by factors of the questioner of childbirth violence.**

| Items | Initial Eigenvalues | | | Extraction Sums of Squared Loadings | | | Rotation Sums of Squared Loadings | | |
|---|---|---|---|---|---|---|---|---|---|
| | Total | % of Variance | Cumulative % | Total | % of Variance | Cumulative % | Total | % of Variance | Cumulative % |
| 1 | 11.137 | 32.756 | 32.756 | 8.2254 | 24.148 | 24.148 | 8.913 | 26.215 | 26.215 |
| 2 | 7.065 | 20.779 | 52.535 | 6.950 | 20.442 | 44.630 | 7.774 | 22.865 | 49.081 |
| 3 | 2.806 | 8.253 | 68.788 | 4.147 | 12.197 | 56.826 | 2.746 | 7.283 | 56.363 |
| 4 | 2.068 | 6.082 | 67.870 | 2.193 | 6.450 | 63.276 | 2.350 | 6.913 | 63.276 |
| 5 | 1.458 | 4.287 | 77.562 | | | | | | |
| 6 | 1.458 | 4.287 | 77.850 | | | | | | |
| 7 | 0.938 | 2.893 | 80.742 | | | | | | |
| 8 | 0.857 | 2.521 | 83.264 | | | | | | |
| 9 | 0.718 | 2.112 | 85.376 | | | | | | |
| 10 | 0.578 | 1.700 | 87.076 | | | | | | |
| 11 | 0.533 | 1.566 | 88.642 | | | | | | |
| 12 | 0.501 | 1.474 | 90.116 | | | | | | |
| 13 | 0.408 | 1.199 | 91.315 | | | | | | |
| 14 | 0.367 | 1.078 | 92.393 | | | | | | |
| 15 | 0.348 | 1.024 | 93.417 | | | | | | |
| 16 | 0.322 | 0.948 | 94.365 | | | | | | |
| 17 | 0.275 | 0.788 | 95.961 | | | | | | |
| 18 | 0.268 | .856 | 95.961 | | | | | | |
| 19 | 0.211 | 0.620 | 97.090 | | | | | | |
| 20 | 0.173 | 0.509 | 97.487 | | | | | | |
| 21 | 0.135 | 0.397 | 97.862 | | | | | | |
| 22 | 0.128 | 0.357 | 98.195 | | | | | | |
| 23 | 0.113 | 0.333 | 98.195 | | | | | | |
| 24 | 0.105 | 0.310 | 98.505 | | | | | | |
| 25 | 0.092 | 0.271 | 98.776 | | | | | | |

Extraction Method: Principal Component Analysis.

The findings regarding the reliability of the instrument are detailed in Table 4. The standard error of measurement was calculated based on the following formula: SEM = SD

$$\sqrt{1 - ICC}$$

## Instrument scoring

The Childbirth Violence Questionnaire (CVQ) is a specialized assessment tool comprising 34 items categorized into four distinct areas. Each item within this questionnaire is framed positively. Respondents select from a Likert scale for each item, with options ranging from never (score 1) to always (score 5), corresponding to rarely (score 2), sometimes (score 3), and often (score 4). The scores for each subscale are derived from the cumulative scores of the items within that sub-scale. The overall score for the tool is obtained by summing the scores of all subscales. The scoring system establishes both minimum and maximum thresholds.

$$\text{Score in percentage} = \frac{\text{Minimum possible raw score} - \text{obtained raw score}}{\text{Minimum possible raw score} - \text{Maximum}} \times 100$$

**Fig 1. The Scree Plot of the questionnaire of the Iranian instrument for assessing violence during childbirth.**

**Table 2. The phases involved in the reduction of items, from the creation of the initial item pool to the completion of the final questionnaire version.**

| Stage | Number of items | Results | Number of remaining items |
|---|---|---|---|
| Initial collection | 108 | During several face-to-face and virtual meetings, along with thorough review and refinement by the research team, 46 comparable items were either consolidated or removed | 65 |
| Quantitative investigation of face validity | 65 | – | 65 |
| Content validity (qualitative) | 65 | A total of 12 items were combined due to overlap, while 2 items were eliminated. | 55 |
| Content validity (quantitative) | 55 | 11 items were removed | 44 |
| Primary reliability | 44 | 3 items were removed | 41 |
| Construct validity | 41 | 7 items were removed | 34 |

Subsequently, a percentage score for each subscale and the entire instrument is computed using a linear transformation formula, as detailed in Table 5.

The higher score obtained indicates a higher level of understanding of childbirth violence. The highest score that a woman can get is 170 and it indicates the highest level of woman understanding violence and the lowest possible score is 34, which indicates the lowest woman's understanding of childbirth violence.

## Floor and ceiling effects

The floor and the ceiling effect were determined for the dimensions and the total instrument. For this purpose, the minimum and maximum possible grades in each field were determined. Then, the frequency of women who had obtained these grades was determined and divided by the total number of samples. The obtained value was reported as a

**Table 3. The results of the final four factor solution of the childbirth violence to the Principal Component Analysis with Varimax rotation and the internal consistency of each factor.**

| Factors name | Items | | Factor Loadings |
|---|---|---|---|
| Psychological and physical abuse | 15 | If I failed to comply with the staff's directives, I was threatened with potential abandonment or denial of care for myself or my infant. | 0.907 |
| | 6 | I experienced judgment based on my physical attributes, including my weight, private areas, and personal grooming. | 0.846 |
| | 5 | I perceived a sense of neglect from the care team for various reasons, including my accent, ethnicity, economic status, and the presence of an infectious disease, particularly in comparison to other women in the ward. | 0.782 |
| | 27 | During my stay, I felt as though my presence was a burden to the staff. | 0.768 |
| | 41 | The hospital's regulations treated me with a strictness that I found unsettling. | 0.731 |
| | 4 | They would prevent me from using the restroom or moving about in the delivery room, without providing any explanation. | 0.731 |
| | 20 | Hearing derogatory remarks, mockery, or shouting undermined my dignity as a human being and as a mother. | 0.700 |
| | 7 | During my hospital stay, I was held responsible for certain complications affecting either myself or my baby, such as issues related to head shape or the birthing process. | 0.677 |
| | 19 | The constant chatter and laughter among the personnel were a source of irritation for me. | 0.663 |
| | 36 | I felt that the quality of my treatment was compromised, leading to a lack of trust in the care I was receiving. | 0.658 |
| | 14 | My companions and I were oblivious to the status of the labor process. | 0.632 |
| | 38 | The inadequacy of facilities, including the absence of specialized departments for mothers and infants, exacerbated my worries. | 0.608 |
| | 34 | I was apprehensive about the potential for neglect due to the limited number of personnel in the maternity ward. | 0.588 |
| | 3 | Throughout my hospitalization, I experienced physical mistreatment, which included actions such as being shoved, struck, or slapped. | 0.588 |
| Non-professional care | 13 | I feel anxious as my inquiries remained unanswered. | 0.714 |
| | 10 | I endured significant discomfort due to the restriction on having a companion during my hospital stay. | 0.714 |
| | 8 | The lack of introductions to my caregivers and unfamiliarity with them heightened my anxiety. | 0.709 |
| | 11 | Their indifference to alleviating my pain, whether from labor or postpartum, was troubling. | 0.705 |
| | 30 | I underwent examinations in a manner that was harsh and inappropriate. | 0.683 |
| | 16 | There was a noticeable disregard for the essential postnatal training, including care for stitches, breastfeeding, and milking the baby. | 0.681 |
| | 23 | During my time in the hospital, my privacy was disregarded. | 0.678 |
| | 17 | I was subjected to distressing discussions regarding childbirth and its implications. | 0.677 |
| | 12 | My basic needs and requests, including hunger, thirst, rest, and sleep, were overlooked. | 0.670 |
| | 22 | I was plagued by the sense that I had become merely a subject for training and practice. | 0.643 |
| | 1 | Numerous medications, including pressure ampoules and others, were administered to me without any explanation provided. | 0.573 |
| delay in offering care | 29 | I sensed that the necessary care I required was being postponed. | 0.795 |
| | 35 | I was apprehensive about the potential for neglect due to the limited number of personnel in the maternity ward. | 0.751 |
| | 32 | Visiting a doctor or midwife was a challenging experience for me. | 0.702 |
| Lack of family support | 44 | During childbirth, the absence of support and companionship from my family and husband left me feeling exhausted and drained. | 0.897 |
| | 43 | During childbirth, my husband and family ignored my emotional and physical conditions. | 0.750 |
| | 42 | My inability to pay the hospitalization fee hindered my access to necessary care and delayed my discharge. | 0.572 |

**Table 4. Reliability with internal consistency methods (Cronbach's alpha), ICC and SEM.**

| Dimensions | Items number | Mean [2] | Cronbach's α pre-test | Cronbach's α post-test | ICC | CI (95%) | SEM |
|---|---|---|---|---|---|---|---|
| Psychological and physical abuse | 14 | 23.13(5.87) | 0.746 | 0.797 | 0.944 | 0.833-0.973 | 1.389 |
| Non-professional care | 14 | 25.56(7.71) | 0.810 | 0.804 | 0.968 | 0.932-0.985 | 1.379 |
| Delay in offering care | 3 | 4.58(1.72) | 0.790 | 0.810 | 0.925 | 0.841-0.964 | 0.471 |
| lack of family support | 3 | 3.93(1.71) | 0.737 | 0.790 | 0.956 | 0.927-0.983 | 0.319 |
| Total | 34 | 57.38(14.57) | 0.900 | 0.889 | 0.950 | 0.892-0.977 | 3.275 |

percentage. The values of floor effect and ceiling effect in all areas and the whole instrument were calculated to be less than 15%. (Table 6).

## Ease of use of the tool

To check the ease of using this tool, the duration of completing the questionnaire was calculated by the participants, the average of which was about 20 minutes and the maximum was 25 minutes. Also, the percentage of women who did not answer each item was determined. In all items, non-response was between 0 and 3%. Therefore, it is simple and understandable to use the tool to measure childbirth violence in mothers.

## Discussion

This study aims to create a valid tool to measure childbirth violence effectively. The constructed questionnaire encompasses the following dimensions: "psychological and physical abuse," "non-professional care," "delay in offering care," and "lack of family support."

The CBVQ encompasses dimensions are supported by previous studies. Based on the results of a study, obstetric violence encompasses mistreatment during childbirth, including physical abuse, emotional harm, and systemic factors [15].

The findings of the present study highlight that childbirth violence is a culturally and socially complex issue, which aligns with previous studies that emphasize the importance of cultural context in understanding women's experiences during childbirth [16,17]. A study highlights that lack of informed consent is a significant issue during childbirth, with inadequate communication and support from health workers. Factors like high stress, workload, and poor organization contribute to ineffective consent practices, necessitating improvements [16].

In fact, the childbirth violence is indeed a culturally and socially complex issue, as it involves ontological disputes over its definition and reflects systemic injustices, particularly in the context of women's rights and the medical authorities're-sponse to citizen demands [17].

The study underscores the necessity for indigenous tools to assess childbirth violence, which is a recurring theme in the literature. Many existing questionnaires focus on respectful maternal care but do not specifically address childbirth violence, highlighting a gap that this study aims to fill. A study aimed to create and validate a German survey tool measuring disrespect and abuse of women during childbirth. The tool includes adapted versions of the Mothers on Respect [3] index, the Mothers' Autonomy in Decision- Making (MADM) scale, a mistreatment index (MIST-I), and items on discrimination experiences in maternity care. The survey, distributed online through social media, had 2045 participants who gave birth in Germany between 2009–2018. Results showed that over 77% experienced mistreatment, notably non-consented care. All scales demonstrated good reliability and validity. The study highlights the significance of addressing disrespect and abuse during childbirth in German maternity care, as it may contribute to post-traumatic symptoms and mental health issues [18]. A study aimed to develop and validate a tool, the "Childbirth Abuse and Respect Evaluation-Maternal Questionnaire" (CARE-MQ), to assess women's perception of abuse or disrespect during childbirth attendance. A multidisciplinary panel created the CARE-MQ, which was tested on 901 Spanish women. The

**Table 5. Calculation of the scores of the constructs of the final questionnaire measuring childbirth violence.**

| Dimensions | Items number | Minimum and maximum scors (Likert scale 1–5) | Score calculation formula |
|---|---|---|---|
| Psychological and physical abuse | 14 | 14-70 | $\dfrac{\text{The raw score obtained} - 14}{56} \times 100$ |
| Non-professional care | 14 | 14-70 | $\dfrac{\text{The raw score obtained} - 14}{56} \times 100$ |
| Delay in offering care | 3 | 3-15 | $\dfrac{\text{The raw score obtained} - 3}{12} \times 100$ |
| lack of family support | 3 | 3-15 | $\dfrac{\text{The} \sim \text{raw score obtained} - 3}{12} \times 100$ |
| Total | 34 | 34-170 | $\dfrac{\text{The} \sim \text{raw score obtained} - 3}{136} \times 100$ |

**Table 6. Determining the effect of the ceiling and floor according to the dimensions of the tool.**

| Dimensions | Items number | Range | Floor effect, % | Ceiling effect, % |
|---|---|---|---|---|
| Psychological and physical abuse | 14 | 14-70 | 5.6 | 0 |
| Non-professional care | 14 | 14-70 | 0 | 0 |
| Delay in offering care | 3 | 3-15 | 4.8 | 0.8 |
| lack of family support | 3 | 3-15 | 11.2 | 0 |
| Total | 34 | 34-170 | 0 | 0 |

questionnaire had good psychometric characteristics and identified four components related to abuse. It showed a negative correlation with the Quality Questionnaire from the Patient's Perspective-Intrapartum and was associated with various childbirth-related variables. CARE-MQ demonstrated good reliability and validity, making it a valuable instrument for evaluating women's experiences of abuse or disrespect during childbirth [19]. Another study aimed to develop an instrument to measure women's mistreatment during childbirth using item response theory in Brazil in 2016. Based on the results of the study, two instruments were created – one for all women (9 items) and one for women in labor (11 items). Results showed a prevalence of mistreatment at 23. 7% and 22% respectively. Key discriminating factors included lack of companionship, not feeling welcome, and not feeling safe [20]. A study aimed to develop reliable tools to measure mistreatment of women during facility-based childbirth based on labor observations. Data from observations of 1,974 women in Nigeria, Ghana, and Guinea were used to develop three measures: an Interpersonal Abuse Scale, an Exams & Procedures Index, and an Unsupportive Birth Environment Index. Factor analysis showed a consistent factor structure for the Interpersonal Abuse Scale in all three countries, with good construct validity and internal reliability [21]. In another study developed and validated a new questionnaire to measure mistreatment of women during childbirth and its impact on satisfaction and perceived quality of care in the West Bank, Palestine. A cross-sectional validation study with 200 postpartum women was conducted, resulting in a questionnaire with three domains: satisfaction of care, perceived quality of care, and experience of mistreatment during childbirth. The questionnaire showed good validity and reliability, with factors identified for each domain and high Cronbach's alpha values [22]. Williams et al, reviewed 37 studies on post-traumatic stress following hospital-based births in several countries. Results showed variation in prevalence estimates based on study design, sample characteristics, measurement tools, and timing [23]. A study on obstetric violence in Quito, Ecuador's health centers found deficiencies in information, accompaniment, and free position options for women during childbirth. Episiotomies and fundal pressure procedures lacked informed consent, while many women were not allowed a chosen companion or birthing position. This highlights the prevalence of obstetric violence in these settings [24].

## Conclusion

Comparing the designed tool with the existing tools, it can be said that the CBVQ is a special tool to measure childbirth violence based on mothers' experiences and understanding of the concept of violence during natural childbirth or emergency cesarean section in public and private centers and semi-private was designed to express violence from the care team, health system, and family. The existing tools mainly emphasize the interpersonal relationship between the care team and the mother. At the same time, the present tool also includes the structures of the health system, including laws, physical environment, facilities, and resources. Also, next to the health system, family support is void as an important sub-scale of the tool for measuring childbirth violence, whose items were precisely extracted from the experiences of mothers and were preserved in factor analysis, showing this important cultural and social issue during childbirth. It is important to understand the violence caused by the lack of social support, which is rooted in cultural weakness and inadequacies. None of the tools related to childbirth violence have a structure with this concept. Therefore, the comprehensiveness and broader coverage of the idea of childbirth violence is one of the features of this tool that makes it different from other tools.

### Strengths

The study started with a qualitative phase, identifying 99 items from participant interviews. This method ensured that the questionnaire reflected real experiences of childbirth violence. Cronbach's alpha showed a high internal consistency score of 0. 956, confirming effective measurement. Exploratory factor analysis with 205 mothers tested construct validity, confirming the instrument's accuracy. Feedback from experts and participants was used to improve the tool's validity and usability.

The choice of experts might lead to bias since their views could be shaped by personal experiences or institutional habits. This may impact what is seen as "essential" items in the questionnaire. The results may be limited to the cultural context of the study, as women's experiences of obstetric violence differ widely across cultures, affecting the relevance of the findings in other settings.

## Supporting information

**S1 Data.**
(RAR)

## Acknowledgments

This study was approved by the Tehran University of Medical Sciences. We thank the coordinators and data collectors who assisted in this study.

## Author contributions

**Conceptualization:** ashraf direkvand-moghadam, Molouk Jaafarpour, Ziba Taghizadeh, Abbas Ebadi, Fatemeh Abbaszadeh.

**Data curation:** ashraf direkvand-moghadam, Molouk Jaafarpour, Fatemeh Najafi.

**Formal analysis:** Ziba Taghizadeh, Abbas Ebadi.

**Methodology:** ashraf direkvand-moghadam, Ziba Taghizadeh, Fatemeh Abbaszadeh.

**Project administration:** ashraf direkvand-moghadam, Molouk Jaafarpour.

**Software:** ashraf direkvand-moghadam.

**Supervision:** Molouk Jaafarpour.

**Validation:** ashraf direkvand-moghadam, Molouk Jaafarpour, Ziba Taghizadeh, Abbas Ebadi, Fatemeh Najafi.

**Visualization:** ashraf direkvand-moghadam, Molouk Jaafarpour, Ziba Taghizadeh, Abbas Ebadi, Fatemeh Najafi.

**Writing – original draft:** ashraf direkvand-moghadam, Molouk Jaafarpour, Ziba Taghizadeh, Abbas Ebadi, Fatemeh Najafi, Fatemeh Abbaszadeh.

**Writing – review & editing:** ashraf direkvand-moghadam, Molouk Jaafarpour, Ziba Taghizadeh, Abbas Ebadi, Fatemeh Najafi, Fatemeh Abbaszadeh.

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
