## [Decision Letter · Decision Letter 0]

31 Jan 2025

PONE-D-24-48647Development and Psychometric Childbirth Violence Questionnaire in Iranian WomenPLOS ONE

Dear Dr. direkvand-moghadam,

Thank you for submitting your manuscript to PLOS ONE. After careful consideration, we feel that it has merit but does not fully meet PLOS ONE’s publication criteria as it currently stands. Therefore, we invite you to submit a revised version of the manuscript that addresses the points raised during the review process.

We look forward to receiving your revised manuscript.

Kind regards,

Othman A. Alfuqaha, Ph.D.

Academic Editor

PLOS ONE

Journal Requirements:

For additional information about PLOS ONE ethical requirements for human subjects research, please refer to http://journals.plos.org/plosone/s/submission-guidelines#loc-human-subjects-research .

Additional Editor Comments:

After careful evaluations, the reviewers have provided detail feedback on your work. Please answer both reviewer's comments and my comments as follows:

Abstract:

-a mixed-methods approach to evaluate. Which mixed method?

-We need sample size and data collection time.

-The CBVQ is a unique tool to assess childbirth violence. Is this sentence correct?

- give us a broader implications.

Introduction

-Your paper needs proofreading by a native English speaker.

- Merge sentence in each other. Paragraphs are too small.

- We need something that we dont know in each paragraph.

- Lack of references are noted in this section.

- Please why your scale is important.

- The aim is not clearly mention. Please re-write it again.

Method are critical and need strong revision. For example, data are too old 2019, Your justification?, study design, answer 5 w questions (When, how, where,.....), unnecessary information and details, 205 participants how?, analysis section is missing.

Results

- Start with your demographic factors.

- Tables are too long and unnecessary. Appendix

- 0.000 is wrong it should be 0.001.

- All results are difficult to read. Please arrange your Tables in 2 Tables for example, mention your results in a scientific way.

Discussion

- Do not repeat your goals, highlighted your findings.

- Discuss your findings with previous studies.

- How we can use this scale globally?

- More international studies and references are needed.

- no need to write strength and limitation in one section.

Please take my comments under consideration, otherwise I will reject the paper.

-

-

Reviewers' comments:

Reviewer's Responses to Questions

**Comments to the Author**

1. Is the manuscript technically sound, and do the data support the conclusions?

Reviewer #1: No

Reviewer #2: Yes

2. Has the statistical analysis been performed appropriately and rigorously? 

Reviewer #1: Yes

Reviewer #2: Yes

3. Have the authors made all data underlying the findings in their manuscript fully available?

Reviewer #1: Yes

Reviewer #2: No

4. Is the manuscript presented in an intelligible fashion and written in standard English?

Reviewer #1: No

Reviewer #2: No

5. Review Comments to the Author

Reviewer #1: 1) Title needs to be changed to include “the assessment of psychometric properties __)

2) PCQ is not introduced, This needs to be introduced before using.

3) Background: does not highlight the magnitude of the problem, especially among Iranian population. Some factual information stated in this section is not supported by references.

4) How 205 mothers were selected for the validation part of the study is not stated. What were the inclusion/exclusion criteria?

5) Some tables are huge and the journal is unable to include then in the manuscript. Authors and editors need to think of a way to make this information available outside the manuscript.

6) It is unclear whether the final version of the questionnaire was made in English language or local language. Also whether authors allow access to the full questionnaire.

7) Although the manuscript is well written, there are some grammatical errors. eg. use of present and past tense

Reviewer #2: 1. The manuscript require professional proof reading.

2. Items' generation procedure must be explained precisely both in the "methodology" and "results" sections.

3. Detailed information is needed in the result section to reflect how the number of identified items reduced from 41 to 34 as stated in page 28.

4. The study limitations were written very superficially and must include a comprehensive list of methodological limitations that might pose bias on the study findings.

6. PLOS authors have the option to publish the peer review history of their article (what does this mean? ). If published, this will include your full peer review and any attached files.

**Do you want your identity to be public for this peer review?** For information about this choice, including consent withdrawal, please see our Privacy Policy .

Reviewer #1: **Yes: ** Sarath Lekamwasam

Reviewer #2: No

---

## [Author Response · Author response to Decision Letter 1]

21 Apr 2025

Dear editor

The authors reviewed the technical points in writing the article point by point

The authors have implemented all suggestions and welcome your valuable guidance in the future

---

## [Decision Letter · Decision Letter 1]

8 May 2025

PONE-D-24-48647R1The assessment of psychometric properties of Childbirth Violence Questionnaire in Iranian WomenPLOS ONE 

Dear Dr. direkvand-moghadam,Thank you for submitting your manuscript to PLOS ONE. After careful consideration, we feel that it has merit but does not fully meet PLOS ONE’s publication criteria as it currently stands. Therefore, we invite you to submit a revised version of the manuscript that addresses the points raised during the review process.

We look forward to receiving your revised manuscript.

Kind regards,

Othman A. Alfuqaha, Ph.D.

Academic Editor

PLOS ONE

Journal Requirements:

Additional Editor Comments :

References with brackets in the text.

Table 1 & 2 are hard to read. I know the items are necessary, but you can let the number only (without words of item) with all results such as CVI, alpha....etc. And make it as supplementary Table. I highly suggest merge Table 1 & 2 without items and with all results needed.

No need for Table of KMO and Bartlett's Test, just write it down.

Also Table 5 & 6 reconsider re-arrange them.

Double check your references.

Reviewers' comments:

Reviewer's Responses to Questions

**Comments to the Author**

1. If the authors have adequately addressed your comments raised in a previous round of review and you feel that this manuscript is now acceptable for publication, you may indicate that here to bypass the “Comments to the Author” section, enter your conflict of interest statement in the “Confidential to Editor” section, and submit your "Accept" recommendation.

Reviewer #1: All comments have been addressed

2. Is the manuscript technically sound, and do the data support the conclusions?

Reviewer #1: Yes

3. Has the statistical analysis been performed appropriately and rigorously? 

Reviewer #1: Yes

4. Have the authors made all data underlying the findings in their manuscript fully available?

Reviewer #1: Yes

5. Is the manuscript presented in an intelligible fashion and written in standard English?

Reviewer #1: Yes

6. Review Comments to the Author

Reviewer #1: Authors have addressed all concerns. They, however, need to see the response of editors regarding the larger tables.

7. PLOS authors have the option to publish the peer review history of their article (what does this mean? ). If published, this will include your full peer review and any attached files.

**Do you want your identity to be public for this peer review?** For information about this choice, including consent withdrawal, please see our Privacy Policy .

Reviewer #1: **Yes: ** Sarath Lekamwasam

---

## [Author Response · Author response to Decision Letter 2]

3 Jul 2025

Dear editor

The authors would like to thank the journal editorial team for their careful consideration of the suggestions and prompt response. Based on editor's opinion, the following changes were made:

References with brackets in the text.

Athours: References were reviewed and organized according to Vancouver Referencing.

Table 1 & 2 are hard to read. I know the items are necessary, but you can let the number only (without words of item) with all results such as CVI, alpha....etc. And make it as supplementary Table. I highly suggest merge Table 1 & 2 without items and with all results needed.

Athours:Tables 1 and 2 were combined, items were removed, and uploaded as a supplemental table.

No need for Table of KMO and Bartlett's Test, just write it down.

Athours: Table of KMO and Bartlett's Test were removed

Also Table 5 & 6 reconsider re-arrange them.

Athours:Table 5 & 6 re-arrange and numbering were changed due to the deletion of the first 3 tables from the text of the article.

Double check your references.

Athours:References were reviewed and organized according to Vancouver Referencing

---

## [Editor Report · Decision Letter 2]

9 Jul 2025

The assessment of psychometric properties of Childbirth Violence Questionnaire in Iranian Women

PONE-D-24-48647R2

Dear Dr.

<table border="0" cellpadding="0" cellspacing="0" class="datatable3" style="border-collapse: collapse; width: 678px; line-height: 14px; color: rgb(0, 0, 51); font-family: verdana, geneva, arial, helvetica, sans-serif; font-size: 11.2px;"> <tbody> <tr style="background-color: rgb(244, 244, 244);"> <td style="padding: 3px; border: 1px solid rgb(255, 255, 255);">ashraf direkvand-moghadam</td> </tr> <tr style="background-color: rgb(244, 244, 244);"> <td style="padding: 3px; border: 1px solid rgb(255, 255, 255); width: 196.094px;"> </td> </tr> </tbody></table>

We’re pleased to inform you that your manuscript has been judged scientifically suitable for publication and will be formally accepted for publication once it meets all outstanding technical requirements.

Kind regards,

Othman A. Alfuqaha, Ph.D.

Academic Editor

PLOS ONE

Additional Editor Comments (optional):

Dear authors, I am satisfied with this revision, and all of my comments have been addressed appropriately. Congratulations and best of luck!
---

## [Editor Report · Acceptance letter]

PONE-D-24-48647R2

PLOS ONE

Dear Dr. direkvand-moghadam,

I'm pleased to inform you that your manuscript has been deemed suitable for publication in PLOS ONE. Congratulations! Your manuscript is now being handed over to our production team.

Kind regards,

on behalf of

Dr. Othman A. Alfuqaha

Academic Editor

PLOS ONE